# Scale Information Enhancement for Few-Shot Object Detection on Remote Sensing Images

Zhenyu Yang [1], Yongxin Zhang [1], Jv Zheng [1], Zhibin Yu [1,2,*] and Bing Zheng [1,2]

1 Faculty of Information Science and Engineering, Ocean University of China, Qingdao 266100, China; yzy7400@stu.ouc.edu.cn (Z.Y.); zhangyongxin@stu.ouc.edu.cn (Y.Z.); zhengju@stu.ouc.edu.cn (J.Z.); bingzh@ouc.edu.cn (B.Z.)
2 Key Laboratory of Ocean Observation and Information of Hainan Province, Sanya Oceanographic Institution, Ocean University of China, Sanya 572024, China
* Correspondence: yuzhibin@ouc.edu.cn

**Abstract:** Recently, deep learning-based object detection techniques have arisen alongside time-consuming training and data collection challenges. Although few-shot learning techniques can boost models with few samples to lighten the training load, these approaches still need to be improved when applied to remote-sensing images. Objects in remote-sensing images are often small with an uncertain scale. An insufficient amount of samples would further aggravate this issue, leading to poor detection performance. This paper proposes a Gaussian-scale enhancement (GSE) strategy and a multi-branch patch-embedding attention aggregation (MPEAA) module for cross-scale few-shot object detection to address this issue. Our model can enrich the scale information of an object and learn better multi-scale features to improve the performance of few-shot object detectors on remote sensing images.

**Keywords:** few-shot object detection; meta learning; remote sensing images; scale information

## 1. Introduction

### 1.1. Background

Thanks to the development of aerospace and sensor technologies, higher-resolution remote sensing images are now available through satellites [1] or unmanned aerial vehicles [2,3]. As remote sensing has great potential in many fields (e.g., disaster monitoring [4–6], environmental monitoring [7,8], urban planning [9,10], and climate observing [11]), object detection on higher-resolution remote sensing images has become increasingly important.

In recent decades of research, object detection has been widely studied in the field of computer vision and has been gradually applied to remote sensing images. In earlier years, researchers attempted to use various detection theories, such as template-matching-based methods [12,13], prior knowledge methods [14], object-based image analysis (OBIA)-based methods [15,16], and machine learning-based methods [17–21] on remote sensing images. Since deep learning technologies have proven their excellent performance in object detection [22,23], people soon adopted these approaches to remote sensing images [20,24] and achieved good results. However, the high performance of deep learning-based models depends on a sufficient amount of training samples [25,26]. Since most remote sensing equipment like airplanes or satellites is expensive, remote sensing images are costly to collect [27]. Labeling small and dense targets on remote sensing images is therefore also challenging [28,29]. Taking advantage of deep learning models with a few remote sensing samples would be a desirable solution.

### 1.2. Related Work

1.2.1. Object Detection on Remote Sensing Images

Before deep learning became widespread, many researchers focused on template-matching-based remote sensing image object detection. Kim, T. et al. [12] used rigid templates to detect some specific objects with simple shapes and small sizes, such as roads. Lin, X. et al. [13] proposed a novel semi-automatic scheme based on template matching and Euclidean distance transform to extract road strips in urban areas. Prior knowledge-based methods are another solution that defines the applicable rules and knowledge for a target of interest. Akcay et al. [14] utilized spectral, structural, and contextual information to detect buildings with complex shapes and roof structures. With the availability and widespread use of sub-meter images, OBIA-based methods [15,16,30,31] have also emerged as a new remote sensing image detection method. The OBIA-based method starts by segmenting the image into uniform regions representing relatively uniform groups of pixels. It then extracts object features from the segments and feeds them to a classifier for classification.

Due to the superior performance of data-based feature extraction [32], many researchers tried to use deep learning-based methods to automatically learn features in the image to detect remote sensing images. Li et al. [20] proposed a multi-angle anchor module based on Faster region-based convolutional neural networks (R-CNNs) [22] to solve the problem of accuracy degradation in rotational change. Gong Cheng et al. [33] proposed an end-to-end cross-scale feature fusion method to obtain a more powerful multi-level feature representation, effectively improving detection accuracy. Zhipeng Deng et al. [34] added an inception module to increase the variety of receptive field sizes in the feature extractor and designed a multi-scale object proposal network that can detect multi-class objects in remote sensing images with large-scale variability. Xiaoliang Qian et al. [35] proposed a multi-level features fusion module to fuse region-of-interest proposals at different levels of feature pyramid networks (FPN) [36], which can make full use of the multi-level features. Unlike the methods mentioned above, which did not consider the few-shot learning cases, our method uses a meta-learning framework to boost the deep learning detector with a few samples.

1.2.2. Few-Shot Object Detection

Few-shot object detection is an essential branch in the field of object detection. An object detection task aims to develop computational models and techniques that simultaneously provide object categories and locations [28]. Faster R-CNN [22] is a classical two-stage detector in object detection. In contrast, single-stage detectors, such as YOLO [23,37], SSD [38], and RetinaNet [39], tend to have faster inference speeds but poorer accuracy performance. A few-shot object detection task is a specific task given only a small number of samples with annotations. Karlinsky, L. [40] proposed the first few sample target detection model called RepMet based on Faster R-CNN. It performs classification by measuring the distance between regions of interest and class representative vectors. Since then, the research on few-shot object detection has attracted extensive attention, especially the method based on meta-learning.

Meta-learning methods, which can learn category-representative vectors through meta-learners and perform feature re-weighting for prediction, have been suggested as one strategy to overcome the few-shot learning challenge [26]. The key idea of meta-learning is to improve the learning ability over time or equivalently. For example, META-YOLO [41] included a meta-feature extractor and max-pooling to obtain the class feature vectors of the support images and then used channel-level multiplication to re-weight the query image and class feature vectors. B. Li et al. [42] introduced a max-margin loss and feature disturbance module in the meta-learner to obtain a more appropriate inter-class distance and to improve the accuracy of detecting new categories. Hu, Hanzhe et al. [43] pointed out that the previous meta-learning network used a global pooling operation to lose detailed local context. They used global attention fusion for feature re-weighting to fully use the

supporting image's information. Guangxing Han et al. [44] first attempted to design a novel Transformer-based backbone network for few-shot object detection.

The transfer learning method [45] is also a solution to the few-shot problem, which mainly relies on fine-tuning from the source dataset to the target dataset and does not require training from scratch. Wang, X. et al. [46] adopted a two-stage fine-tuning strategy, which achieved good detection accuracy by freezing the backbone network weights and only fine-tuning the last layer of the detector head during the fine-tuning stage. Jiaxi Wu et al. [47] proposed a positive sample enhancement strategy to solve the scale sparsity problem. Qiao, Limeng et al. [48] proposed a simple fine-tuning-based network named Decoupled Faster R-CNN (DeFRCN), which alleviates the problem of data scarcity by adding multi-task decoupling and multi-stage decoupling modules. Sun, Bo et al. [49] proposed few-shot object detection via contrastive proposal encoding. This method can achieve more robust object representations through contrastive learning, which helps to improve the classification accuracy of detected objects.

The above methods were designed for datasets with natural scenes. In contrast, remote sensing datasets have dense and small targets with a wide range of size variations. This paper focuses on the cross-scale and the problem of a sparse distribution of scale information for remote sensing object detection.

### 1.2.3. Few-Shot Object Detection on Remote Sensing Images

Considering the characteristics of remote sensing datasets, researchers have proposed tentative designs for few-shot object detection algorithms and achieved remarkable results. Li, Xiang et al. [50] proposed the first few-shot object detection algorithm on remote sensing images, adding a multi-scale architecture based on META-YOLO to adapt to the scale change in remote sensing objects. Zhao, Zhitao et al. [51] proposed a path-aggregation module to aggregate features from all feature levels and to alleviate the problem of scale sparsity. Wang, Yan et al. [52] designed a context information refinement module to learn discriminative context features, which can detect objects of different scales and among clutter such as in complex backgrounds. Zhang Yuchen et al. [29] proposed an adaptive global similarity module, which introduces spatial information and computes the global similarity between support instances and query instances to alleviate the problem of significant appearance variations in the same category.

Most of these methods used a multi-layer pyramid structure to produce semantic feature maps of different sizes to enrich the scale space of features. However, they do not change the scale distribution of the data, which is crucial to few-shot object detection in remote sensing images. To address this issue, we propose a framework to enrich the scale information of the data by constructing a scale space for the targets.

### 1.3. Problems and Contributions

This paper studies the scale information distribution of remote sensing images. When the amount of data used for training is limited, the scale information of remote sensing images is insufficient. Figure 1 shows an example. We randomly chose n ($n = 5, 10, 20$) images and counted the scale distribution of the DIOR remote sensing dataset [28].The red columns represent the scale distribution of a sufficient amount of data, and the blue ones represent the scale distribution of a small number of randomly selected samples. There is a vast difference between the scale distribution of the five randomly selected samples and the case with sufficient data. As the number of instances increases, the difference between the two distributions gradually decreases. A typical case in remote sensing images is that the scale information for many instances in one image is similar (Figure 1c). The lack of scale variety affects the performance of the detector. Furthermore, cross-scale is a common problem in object detection of remote sensing images. Despite extensive studies in previous work, this remains a challenging problem [36,47,50] in few-shot object detection tasks.

To address these problems, we propose a few-shot object detection method on remote sensing images, design a Gaussian-scale enhancement strategy, and add a multi-scale

attention module. To tackle the problem of sparse scale distribution with few training samples, we proposed the Gaussian-scale enhancement strategy, which can enrich the scale information of the target. In the MPEAA module, we use multi-scale patch embedding to obtain the feature map of different receptive fields and calculate the attention. Furthermore, the multi-path aggregation method we designed can learn better multi-scale features.

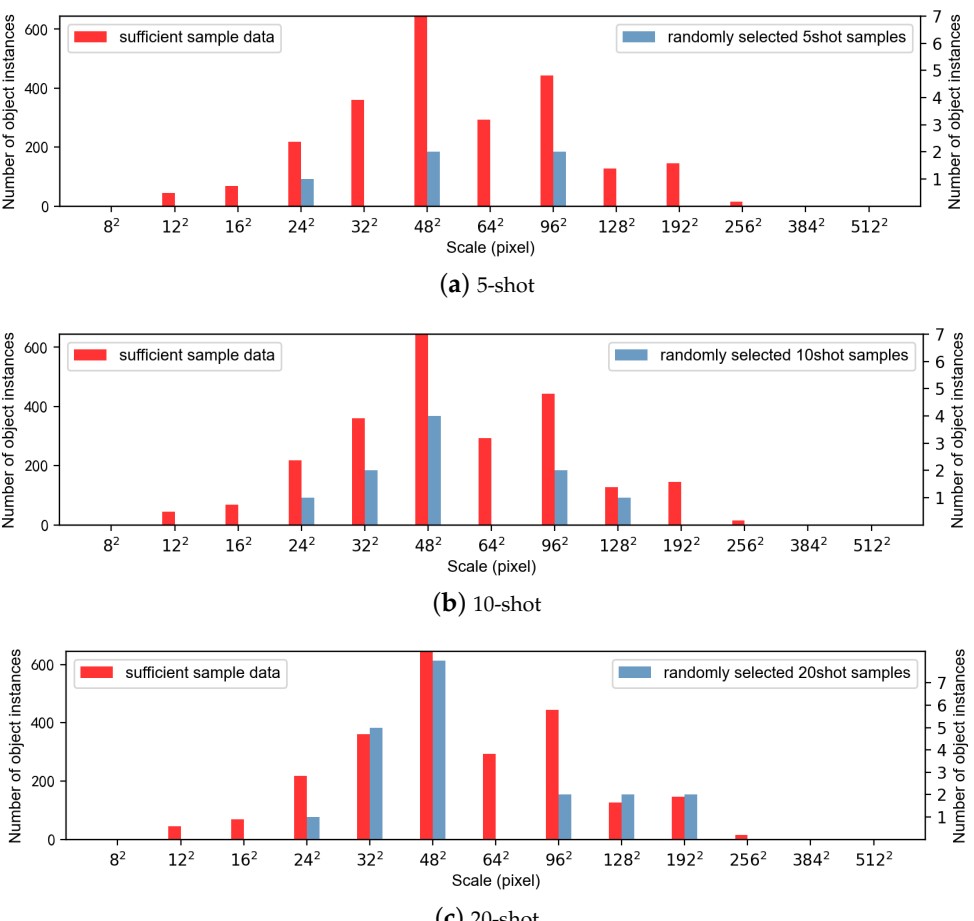

**Figure 1.** An example of scale distribution of windmills in the DIOR dataset under a few shots [28]. The red columns and left axes indicate the scale distribution for sufficient samples, while the blue ones and the right axes describe the cases when the sample number is insufficient. (**a**) The distribution comparison under five shots from three randomly selected images; (**b**) the distribution comparison under ten shots from five randomly selected images; (**c**) the distribution comparison under twenty shots from nine randomly selected images.

Our main points of contribution are as follows:

- We propose a Gaussian-scale enhancement strategy to enrich scale information with limited training data. The strategy can enrich the scale information of a target by constructing a Gaussian-scale space for a small sample of targets and can feed it into the network for training to improve the performance of detectors.
- We propose a multi-branch patch-embedding attention aggregation module with meta-learning. We use multi-scale patch embedding to represent fine and coarse features effectively. Multi-branches perform feature re-weighting based on Transformer attention. The resulting features are aggregated, which can better learn multi-scale features to alleviate the cross-scale problem and improve the performance of detecting new classes of targets.

- We experimentally demonstrate the improved effectiveness of our method on the DIOR [28], NWPU VHR-10 [53–55], and DOTA [56–58] datasets. Our method outperforms the state-of-the-art methods on both datasets.

## 2. Methods

Our model adopts a meta-learning-based architecture [41–44] for few-shot object detection. Similarly to other work, our method contains the support and query branches. The support branch is responsible for extracting the class information, and the query branch is responsible for extracting the features of the detected images. Following the definition of meta-learning [41], we re-weighted the query features with the support features and then fed the results into the subsequent detection head network. As mentioned in Section 1, the object-scale variety is affected by fewer samples. Thus, we propose a Gaussian-scale enhancement strategy which uses Gaussian convolution to enrich the scale information of supporting images.

Query features use three different sizes of patch embeddings for encoding. The output of the query branch features patch embeddings of different sizes. We used the self-attention method in the Transformer for re-weighting features and then fused the multi-branch outputs in a pyramidal manner to obtain a feature map containing multi-scale information. The feature maps were fed into the RPN network to generate proposal boxes related to the supported feature categories. Then, we used region of interest (RoI) align [59] to crop out the feature proposal regions and to fine-tune the regression box in the second stage to obtain the detection result. Our loss function mainly consists of RPN loss and head loss [22]. The RPN loss and head loss include classification loss and regression loss. Our overall architecture is shown in Figure 2. In the following subsections, we first introduce the research background, including the problem definition of few-shot object detection and the construction process of the scale space in Section 2.2. Next, we illustrate the Gaussian-scale enhancement strategy and the multi-branch patch-embedding module based on the Transformer architecture design in Sections 2.2 and 2.3.

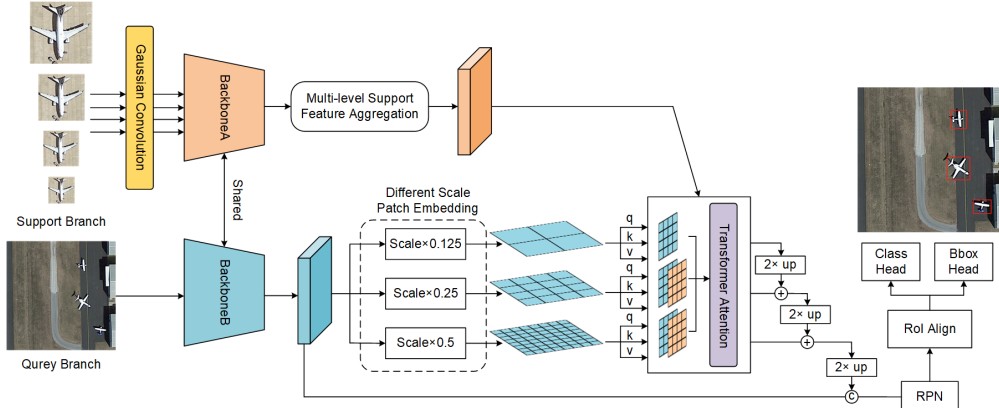

**Figure 2.** Overview of the proposed method. The method inherits a meta-learning framework [41–44] with a two-branch structure. We applied a Gaussian-scale enhancement (GSE) strategy to the image categories in the support branch. The role of the Gaussian Convolution module is to construct a Gaussian-scale space and to enrich the scale information on the support image. Then, we used a multi-branch patch-embedding attention aggregation (MPEAA) module for feature re-weighting. Finally, we fed the re-weighted features to the region proposal network (RPN) network and the detection head to obtain the object categories and locations.

### 2.1. Research Background

First, we explain the problem setting of few-shot object detection [50]. Few-shot target detection means learning a detection model from a base class dataset that can detect targets of a new class. The few-shot target detection tasks usually require only a small number of annotated new class samples. There are sufficient samples for model training for base class

targets to gain prior knowledge, while new class targets have only limited samples. Few-shot object detection can learn meta-knowledge from the base class dataset and apply it generously to new class targets. This setting is standard in real-world application scenarios where a large-scale dataset covers a limited number of classes. When a new class of targets needs to be detected, it is costly to re-collect a large amount of data for training.

We followed the general settings for few-shot object detection on remote sensing images [50,52]. The dataset was divided into two parts by category, one containing the base category $C_{\text{base}}$ with a large amount of data and the other containing the novel category $C_{\text{novel}}$ with a small amount of sample data. There was no intersection between $C_{\text{base}}$ and $C_{\text{novel}}$. Let us define $I$ as an image and $Y = \{(b_n, y_n)\}^N$ as a list of $N$ objects in $I$. $b_n \in \mathbb{R}^4$ is the bounding box of the $n_{th}$ instance, and $y_n \in \{0, 1\}^{|C_{\text{base}} \cup C_{\text{novel}}|}$ denotes an associated one-hot label encoding. $(I, Y)$ is a sample in $D_{\text{base}} \cup D_{\text{novel}}$.

In accordance with meta-learning-based object detection [50,52], we constructed several episodes as input to the model. Each episode consisted of a set of supporting images and a query image. The supporting images contained K annotated objects per class for the K-shot few-shot object detection setting. The training process of few-shot object detection contained two steps. First, we constructed episodes with $C_{\text{base}}$ to train the network and learned prior knowledge from $C_{\text{base}}$. Second, we used several episodes composed of $C_{\text{base}}$ and $C_{\text{novel}}$ to fine-tune the network so that the network can detect novel categories.

Scale-space theory [60] is a classic theory in computer vision. The scale space represents the scale range of an object. Koenderink [60] and Lindeberg [61] have shown that, under various reasonable assumptions, the Gaussian function is the best candidate for a scale-space kernel [62]. Following scale-space theory, we can construct a scale space by convolving the target object image with a series of Gaussian filters with different parameters. This operation can produce images at different scales, efficiently supporting the target object's scale variety. The calculation process is as follows:

$$g_\sigma(x, y) = \frac{1}{2\pi\sigma^2} e^{-(x^2+y^2)/2\sigma^2} \tag{1}$$

$$f_\sigma(x, y) = g_\sigma(x, y) * f(x, y) \tag{2}$$

where $x, y$ denote the pixel positions of the 2D image; $\sigma$ is the scale parameter in terms of the standard deviation of the 2D Gaussian filter; $f(x, y)$ denotes a 2D image; $g_\sigma(x, y)$ denotes a Gaussian filter; and $*$ represents a convolution operation.

Constructing a scale space for a certain object can enrich its scale information without affecting the texture and shape information. The scale-invariant feature [63] is defined as an unchangeable feature even though its object-scale changes [64]. Scale-invariant feature transform (SIFT) [62] extracts scale-invariant features from a scale space generated by Gaussian filters. SIFT can perform key point matching well, which illustrates that a scale space image generated by the Gaussian filter will retain key information about the target. Following this idea, we used Gaussian kernels to construct a scale space to train our meta-learning object detection network and to enrich the scale information while retaining the texture and shape information of the target objects.

### 2.2. Gaussian-Scale Enhancement Strategy

Inspired by the scale-space theory [60,63], we designed a Gaussian-scale enhancement strategy that can be applied to support branch learning. First, we cropped various objects through an annotation frame and resized the cropped image into four sizes $\{64^2, 128^2, 256^2, 512^2\}$. Then, we used Gaussian convolution kernels of different scale parameters $\sigma$ to perform convolution on these four scaled images.

$$f_{\sigma_i}(x, y) = g_{\sigma_i}(x, y) * f(x, y) \tag{3}$$

$$\sigma = 0.3 * ((ksize - 1) * 0.5 - 1) + 0.8 \qquad (4)$$

Image processing library OpenCV provides Formula (4) for the adaptive calculation of $\sigma$. *ksize* represents the size of the convolution kernel, which is taken as odd numbers. We take *ksize* with 3, 5, 7, and 9, respectively, and obtain the value of $\sigma_i = \{0.8, 1.1, 1.4, 1.7\}$.

The results obtained with the original image construct a scale space for this class of targets, as shown in Figure 3. After obtaining images of object at various scales in the scale space, we inputted them into the network for training. Augmenting the new category scale distribution, we obtained support image representations with richer scale information.

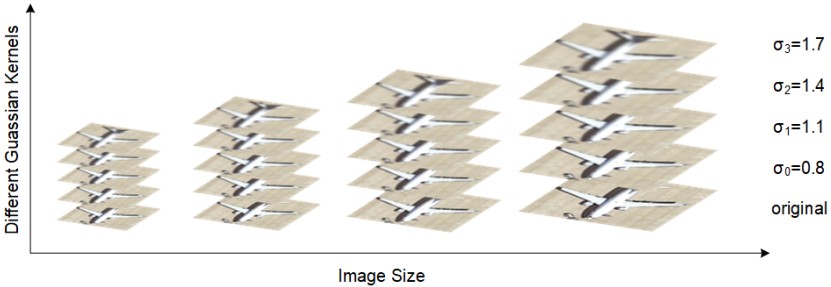

**Figure 3.** A Gaussian-scale space constructed for a specific instance by performing a resize operation, followed by Gaussian filtering of different intensities.

Next, we randomly selected one image at each size in the entire scale space as the input to the network support branch. Since the inputs contain different sizes, we designed a multi-level support feature aggregation (MSFA) module for scale fusion, as shown in Figure 4. The support images were input into the backbone network, which had the multi-scale feature maps of FPN $\{P2, P3, P4, P5, P6\}$. For the given support images of different sizes $\{S_1 : 64^2, S_2 : 128^2, S_3 : 256^2, S_4 : 512^2\}$, we selected multi-scale feature maps of FPN as the output $\{P_2(S_1), P_3(S_2), P_4(S_3), P_5(S_4)\}$. The four feature maps had the same size. We concatenated (Concat) them and then used two convolutional layers for feature fusion.

$$X_s = Conv(Conv(Concat(P_2(S_1), P_3(S_2), P_4(S_3), P_5(S_4)))) \qquad (5)$$

where $Conv()$ denotes a standard convolution operation in which the convolution kernel size is $1 \times 1$. Furthermore, $Concat()$ represents the channel-level concatenation. We can obtain the final support class representatives $X_s$ through Equation (5).

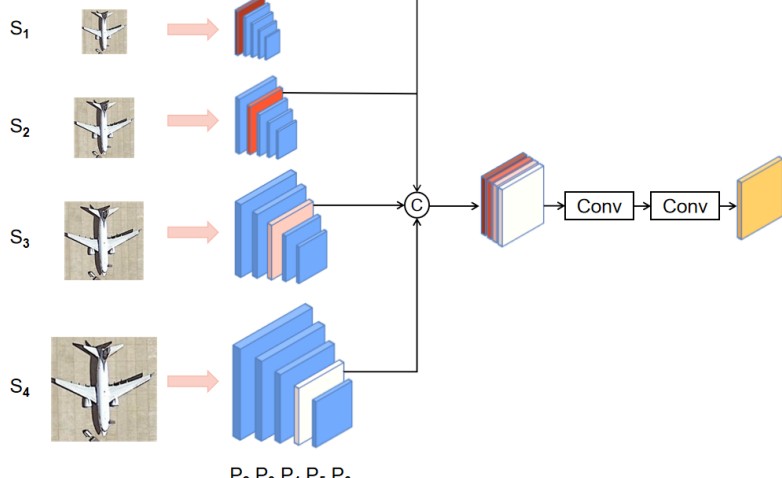

**Figure 4.** Detail of the multi-level support feature aggregation module (MSFA). The support images resized to various scales. Specific feature maps from FPN were selected for fusion. We used Concat and two convolutional layer to aggregate feature maps.

Although the GSE strategy we proposed cannot make the scale information completely consistent with the real distribution, it can increase scale information and alleviate the problem of sparse scale distribution.

### 2.3. Multi-Branch Patch-Embedding Attention Aggregation

Previous few-shot object detection work [41,42,50] mainly used support features after global pooling to modulate query features, which is prone to losing detailed local context information [43]. Therefore, some researchers [43,44] have attempted to match query and support features at the pixel level without using global average pooling operations. They used the cross-attention mechanism to calculate the pixel-level similarity between query features and support features, which can be seen as an extension of the self-attention mechanism in Transformer. However, since these methods did not consider the cross-scale problem, applying these methods to remote-sensing images would be challenging.

Enlightened by the idea of multi-scale patch embeddings proposed in MPViT [65] and CrossViT [66], we designed a multi-branch patch-embedding attention aggregation module for feature re-weighting, which can learn better multi-scale features. Vision Transformer's (ViT) [67,68] primary process generally divides the feature map into several patches. Each patch is flattened and goes through a trainable linear projection. The output of this projection can be seen as the patch embeddings. After linear projection, we mapped the input patch sequence to query $Q$, key $K$, and value $V$ with learnable matrix $W_Q, W_K, W_V$ and then used Formula (7) to calculate the attention.

$$Q = XW_Q, K = XW_K, V = XW_V \tag{6}$$

$$Attention(Q, K, V) = softmax\left(\frac{QK^T}{\sqrt{d_k}}\right)V \tag{7}$$

Overlapping patch embedding is proposed in PVTV2 [69]. Ref. [69] proves that overlapping patch embedding performs better on classification and detection tasks than original patch embedding because it can model the local continuity of images and feature maps via the overlapping sliding window. So, we use overlapping patch embedding to divide the images into several patches on the MPEAA module. We used three parallel patch-embedding layers with different patch sizes to obtain the multi-scale projection of the query feature.

$$X_q^i = PE_i(X_q) \tag{8}$$

where $X_q$ represents the query feature and $PE_i()$ represents the patch-embedding layers. Patch-embedding layers use convolution operations with different kernel sizes and strides to project multi-size patches.

After the patch-embedding layer, the input feature $X \in \mathbb{R}^{H \times W \times C}$ will map to the $C_i$ dimension. Additionally, the output $PE_i(X) \in \mathbb{R}^{H_i \times W_i \times C_i}$ has the following height and width:

$$H_i = \left\lfloor \frac{H - k_i + 2p_i}{s_i} + 1 \right\rfloor, W_i = \left\lfloor \frac{W - k_i + 2p_i}{s_i} + 1 \right\rfloor \tag{9}$$

where $k$ represents the kernel size of the patch-embedding layers, $s$ represents the stride size, and $p$ represents the padding size. The parameter settings of each $PE()$ layer are shown in Table 1.

**Table 1.** Parameter settings for each $PE()$ layer.

| $PE_i()$ | $k$ | $s$ | $p$ |
|---|---|---|---|
| $i = 0$ | 3 | 2 | 1 |
| $i = 1$ | 7 | 4 | 3 |
| $i = 2$ | 15 | 8 | 5 |

Then, we mapped each branch's query patch sequence $X_q^i$ to $Q_q^i$, $K_q^i$ and $V_q^i$. Furthermore, we mapped the support feature $X_s$ to $K_s^i$ and $V_s^i$. Next, we concatenated the K-V pairs of the two branches to enable the deep interaction of support and query features. Subsequently, we computed the attention with aggregated features.

$$Y_i = Attention(Q_q^i, Concat(K_q^i, K_s^i), Concat(V_q^i, V_s^i)) \qquad (10)$$

where $Y_i$ represents the results of the three parallel branches with different sizes. We upsampled lowest-resolution feature maps using nearest neighbor upsampling by a factor of 2. Then, the upsampled feature map was merged with the following branch output by element-by-element addition, and this process was repeated until we obtained the highest-resolution feature map. This aggregation process was iterated until the highest-resolution feature map was generated. Finally, the result was concatenated with query feature $X_q$ and convolution operations for learnable feature aggregation.

$$Y_{final} = Conv(Concat(X_q, UP(Y_0 + UP(Y_1 + UP(Y_2))))) \qquad (11)$$

where $+$ represents an element-wise addition and $UP()$ represents nearest neighbor upsampling. We adopt the method of gradually upsampling and combining features of different scales, which can integrate multi-scale features well. We additionally concatenate multibranch fusion features and the query feature to keep the information intact. Then, the final aggregation results $Y_{final}$ will be fed into the RPN network to generate the RoI. Finally, after RoI align is performed, these RoIs perform the final classification and regression tasks.

## 3. Experiments and Results

### 3.1. Datasets

The NWPU VHR-10 [53–55] dataset contains 800 high-resolution satellite images collected from Google Earth and Vaihingen datasets and then manually annotated by experts. The dataset contains ten categories (airplanes, ships, storage tanks, baseball fields, tennis courts, basketball courts, ground runways, ports, bridges, and vehicles).

The DIOR dataset [28] is a large-scale benchmark dataset for object detection in optical remote sensing images. The DIOR dataset is satellite images from Google Earth and containing 23,463 images and 192,472 instances, covering 20 object classes. The 20 object classes are airplane, airport, baseball field, basketball court, bridge, chimney, dam, expressway service area, expressway toll station, harbor, golf course, ground track field, overpass, ship, stadium, storage-tank, tennis court, train station, vehicle, and windmill.

The "Dataset for Object Detection in Aerial Images" (DOTA) dataset is specifically designed for detecting and classifying various object categories within high-resolution aerial images. Aerial imagery is collected using aircraft or drones equipped with high-resolution cameras. Currently, there are multiple versions of DOTA datasets. This paper uses the DOTAv1.0 version for the study of few-shot object detection. The DOTA dataset contains 2806 images of 15 classes. The object classes include planes, ships, storage tanks, baseball diamonds, tennis courts, basketball courts, ground track fields, harbors, bridges, large vehicles, small vehicles, helicopters, roundabouts, soccer ball fields and swimming pools.

Our experimental setup uses the same general settings as previous work [29,50–52]. For the NWPU VHR-10 dataset, which is also based on a satellite, three classes (airplane, baseball diamond, and tennis court) were used as novel classes, and the others were used as base classes. For larger datasets such as the DIOR dataset, five classes (airplane, baseball field, train station, windmill, and tennis court) were used as novel classes and the others were used as base classes. For the aerial dataset DOTA, we selected four novel classes (plane, baseball-field, tennis-court and helicopter). When training the base class with sufficient sample instances, the few-shot object detection focuses only on the performance of the novel class target. K sample instances can be used per class when training the fewer new class samples. For the NWPU VHR-10 dataset, the number of annotated instances $K$ was set as 3, 5, and 10. For the DIOR dataset, the number of annotated instances $K$ was set

as 5, 10, and 20. For the DOTA dataset, the number of annotated instances *K* was set as 3, 5, 10, and 20. We used ten random number seeds and calculated their average to ensure that we could obtain algorithmically stable results.

### 3.2. Experimental Set and Evaluation Metrics

We trained the model using two RTX 3090 GPUs, each with a batch size of 4. The base training phase used a stochastic gradient descent (SGD) optimization algorithm with an initial learning rate of 0.01. We trained 10,000 iterations on the NWPU VHR-10 dataset, dividing the learning rate by ten into 4000 and 6000 iterations. We also used the same initial learning rate in the fine-tuning phase. We trained 20,000 iterations on the base class for the DIOR data and performed learning rate decay from 14,000 to 16,000 iterations. Training the model on both datasets for up to 2000 iterations is sufficient to achieve good performance.

We use the mean average precision (mAP) [70] as an evaluation metric to evaluate the detection performance (Equation (12)). The mAP is a commonly used evaluation metric in object detection. It comprehensively considers precision and recall to evaluate detection results. Precision measures the accuracy of the predicted bounding boxes, while recall measures the fraction of ground-truth objects that are successfully detected. We can obtain the AP value of the c-th category by calculating the area under the precision–recall (P–R) curve, the formula for which is shown in Equation (12).

$$AP_c = \int_0^1 P_c(R)dR \tag{12}$$

where *P* denotes precision, *R* denotes recall, and *mAP* refers to the average AP value of multiple detection categories.

$$mAP = \frac{\sum_{c=1}^{n} AP_c}{n} \tag{13}$$

where *n* denotes the number of novel classes under the few-shot object detection setting.

### 3.3. Results

We validated our proposed few-shot detector on the NWPU VHR-10 and DIOR datasets along with state-of-the-art FSOD methods, such as meta-learning-based Rep-Met [40], FSODM [50], PAMS-Det [51], DCNet [43], FCT [44], and fine-tuning-based CIR-FSD [52], MPSR [47], TFA [46], SAGS [29], and DeFRCN [48]. The meta-learning-based methods construct several episodes to learn and obtain the class representative vector. These methods then use the class representative vector to re-weight the query features and to predict the target. The transfer-learning-based methods use a batch training strategy and then directly predict the target in the image.

The DIOR dataset is a large-scale dataset. Table 2 shows that our proposed method achieves the best performance. We highlight the mAP score of the best detection performance in bold for Tables 2–7. Compared with the baseline model DCNet, our method has a 1.5% improvement in the 20-shot setting, a 2.3% improvement in the 10-shot setting, and a 2% improvement in the 5-shot setting.

As shown in Figure 5a, the airplanes in the pictures are of different sizes and environments, which makes it a challenging detection task. With the help of the Gaussian-scale enhancement strategy, our method has the best performance with the lowest missed detection rate. The field rotation in Figure 5b shows another challenge. Since the proposed MPEAA module can efficiently extract the multi-scale features, the proposed method achieves the best performance in Figure 5b. Furthermore, when targets with tiny or large sizes are in the scene, such as the tennis court in Figure 5c, our method can also accurately localize and classify them.

**Table 2.** Experimental results for novel class detection on the DIOR dataset.

| Method | 5-Shot | 10-Shot | 20-Shot |
|---|---|---|---|
| RepMet [40] | 8 | 14 | 16 |
| TFA [46] | 25 | 31 | 37 |
| FSODM [50] | 25 | 32 | 36 |
| PAMS-Det [51] | 33 | 38 | - |
| SAGS [29] | 34 | 37 | 42 |
| CIR-FSD [52] | 33 | 38 | 43 |
| DeFRCN [48] | 27.2 | 31.7 | 34.8 |
| FCT [44] | 32.3 | 34.6 | 42.4 |
| MPSR [47] | 32.9 | 38.4 | 44.2 |
| DCNet [43] | 33.2 | 38.3 | 44.9 |
| Ours | **35.2** | **40.6** | **46.4** |

The results of the NWPU VHR-10 dataset are shown in Table 3. Our method outperforms the other state-of-the-art methods, with a 3.4% improvement in the 3-shot setting, a 3.3% improvement in the 5-shot setting, and a 2.6% improvement in the 10-shot setting when compared with DCNet.

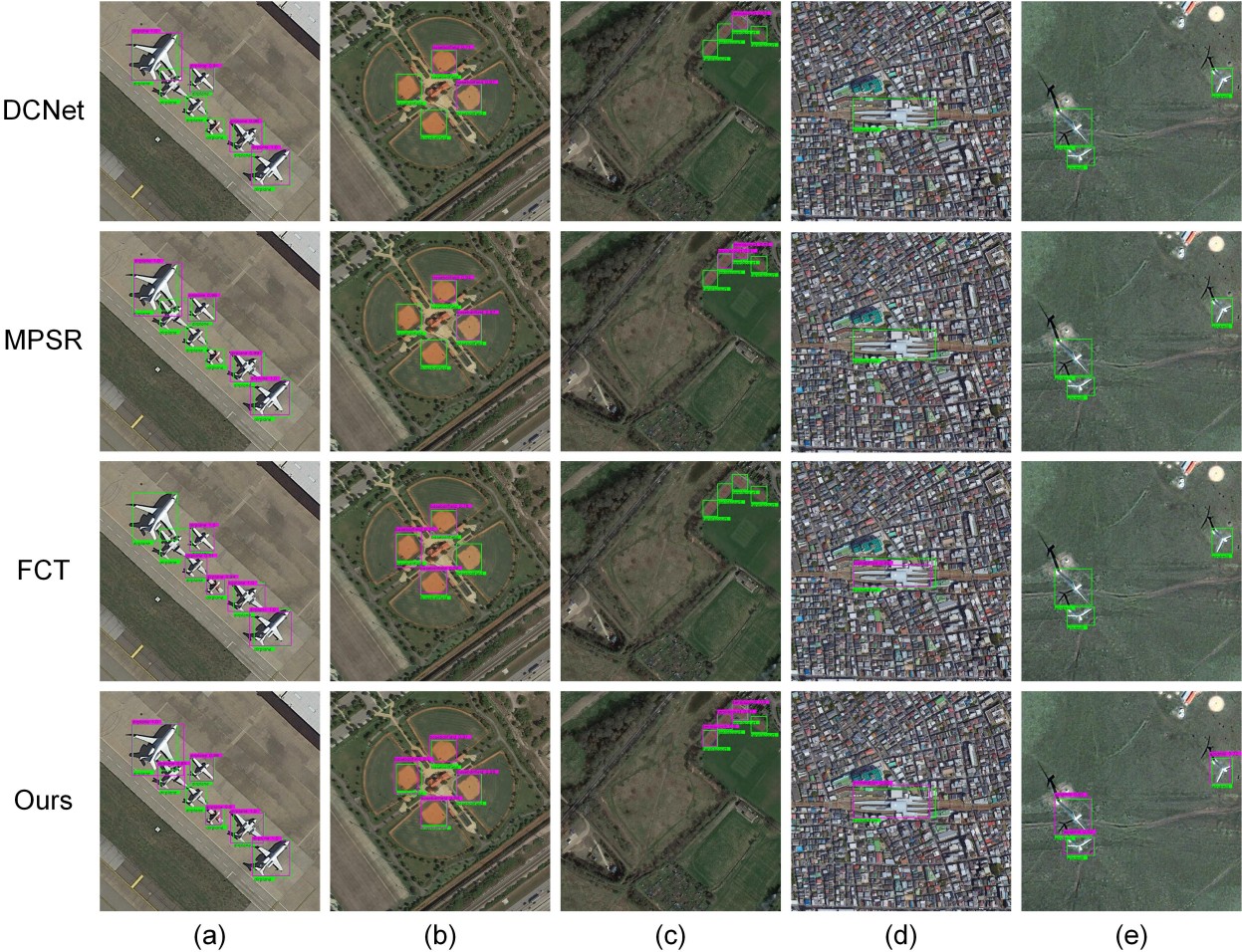

**Figure 5.** Visualization of object detection on the DIOR dataset. The green bounding box represents a ground truth bounding box and the purple bounding box represents the model's detection results: (**a**) airplane, (**b**) baseball field, (**c**) tennis court, (**d**) train station, and (**e**) windmill.

**Table 3.** Experimental results for novel class detection on the NWPU VHR-10 dataset.

| Method | 3-Shot | 5-Shot | 10-Shot |
|---|---|---|---|
| RepMet [40] | 23 | 23 | 26 |
| TFA [46] | 29 | 49 | 65 |
| FSODM [50] | 32 | 53 | 65 |
| PAMS-Det [51] | 37 | 55 | 66 |
| SAGS [29] | 51 | 66 | 72 |
| CIR-FSD [52] | 54 | 64 | 70 |
| DeFRCN [48] | 33.5 | 41.9 | 50.7 |
| FCT [44] | 54.6 | 58.8 | 63.8 |
| MPSR [47] | 59.3 | 65.9 | 69.5 |
| DCNet [43] | 56.7 | 64.8 | 72.1 |
| Ours | **60.1** | **68.1** | **74.7** |

Figure 6 shows a visualization of the experimental results on the NWPU-NHR10 dataset. In Figure 6a, the missed detection rate is relatively high when the object (aircraft) size changes. Aided by the Gaussian-scale enhancement strategy, our method has the best detection performance. Figure 6c shows a dense object detection challenge: almost all the tennis courts are small and densely arranged. While most other methods include many missing objects, our method can accurately detect densely arranged small-sized targets with the help of the MEPAA module.

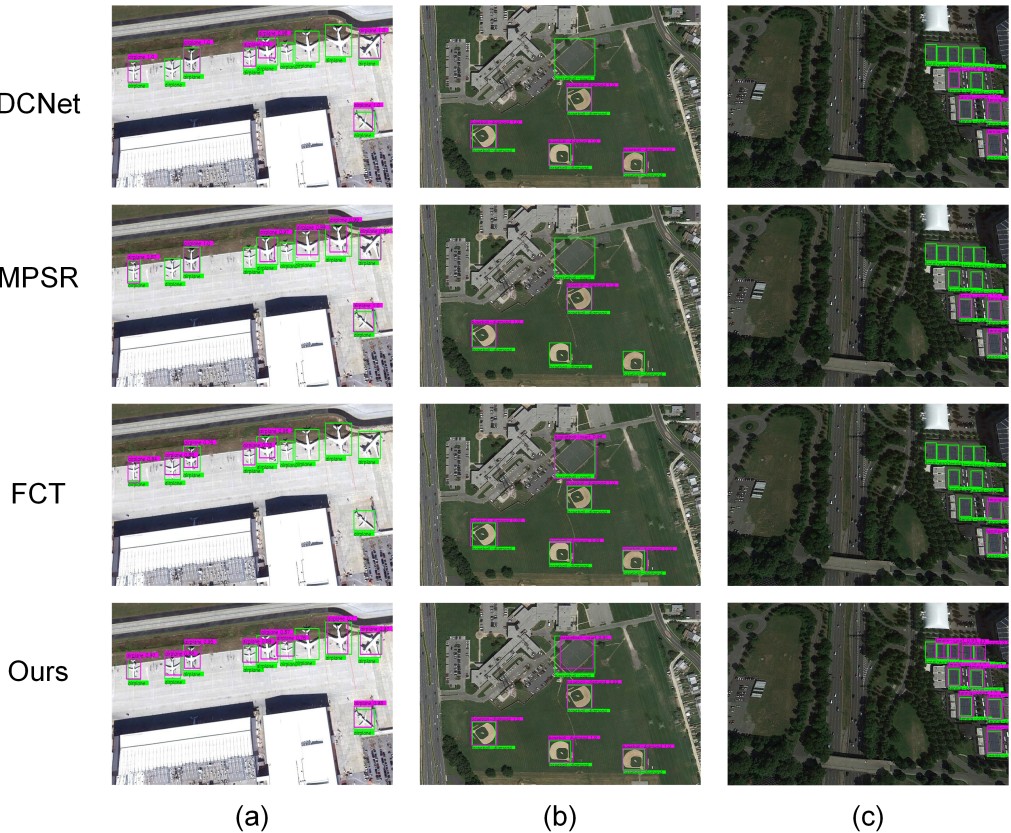

**Figure 6.** Visualization of object detection on the NWPU-NHR10 dataset: (**a**) airplane, (**b**) baseball field, and (**c**) tennis court.

Table 4 compares our method with other few-shot object detectors. Our method consistently outperforms DCNet and other counterparts with each shot setting. Compared to MPSR, our method significantly improves accuracy by 1.7%/3.8%/2.9%/3.6% mAP

with 3/5/10/20-shot setting. The detection result verifies the effectiveness of our proposed method on satellite and aerial image data.

**Table 4.** Experimental results for novel class detection on the DOTA dataset.

| Method | 3-Shot | 5-Shot | 10-Shot | 20-Shot |
|---|---|---|---|---|
| CIR-FSD [52] | 16.7 | 20.4 | 24.3 | 27.9 |
| DeFRCN [48] | 13.8 | 23.8 | 31.2 | 34.6 |
| MPSR [47] | 21.6 | 29.7 | 41.1 | 50.1 |
| DCNet [43] | 21.9 | 29.2 | 35.4 | 48.1 |
| Ours | **23.3** | **33.5** | **44.0** | **53.7** |

Figure 7 shows a visualization of the experimental results on the DOTA dataset. With the help of the GSE strategy, our method can successfully detect planes and baseball fields of various sizes, and other methods will lose some targets. Our proposed MPEAA module can achieve high-quality feature reweighting, and our method can detect densely arranged tennis courts. The red box represents a misdetection. Figure 7d shows a challenge case. The helicopter can be considered as a class with similar patterns of planes, causing both MPSR and DCNet to detect helicopters as airplanes. The proposed model can successfully recognize the helicopter with the help of MPEAA module.

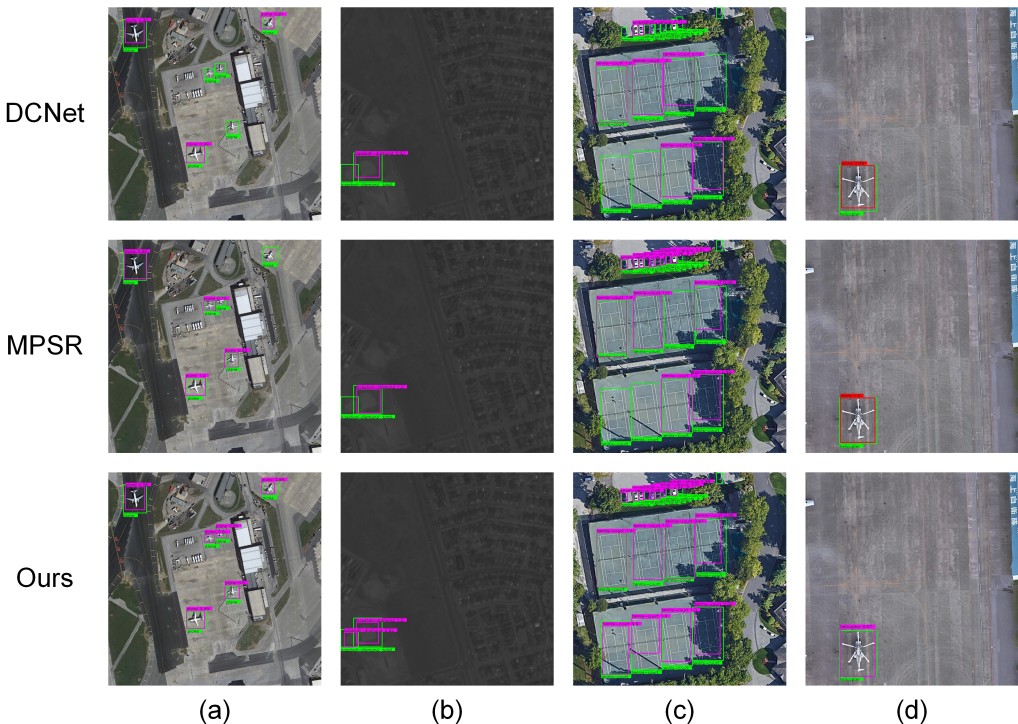

**Figure 7.** Visualization of object detection on the DOTA dataset: (**a**) plane, (**b**) baseball field, (**c**) tennis court, and (**d**) helicopter.

### 3.4. Ablation Experiments

To demonstrate the effectiveness of each module of our proposed method, we conducted ablation experiments using the Gaussian-scale enhancement strategy and the multi-branch patch-embedding attention aggregation module. The ablation experiment Table 5 shows that GSDA achieved a 0.7–1.7% improvement on the DIOR dataset. As shown in Table 6, an improvement of 1.5–3% is also obtained on the NWPU VHR-10 dataset, proving the strategy's effectiveness. Also, using the MPEAA module, improvements of about 0.7% and 0.4–0.8% were obtained on the DIOR dataset and the NWPU VHR-10 dataset,

respectively. We used the Gaussian-scale enhancement strategy and the MPEAA module to maximize these improvements.

**Table 5.** Ablation experiment on the DIOR dataset.

| GSE | MPEAA | 5-Shot | 10-Shot | 20-Shot |
|:---:|:---:|:---:|:---:|:---:|
|  |  | 33.2 | 38.3 | 44.9 |
| √ |  | 34.9 | 39.9 | 45.6 |
|  | √ | 33.9 | 39.0 | 45.6 |
| √ | √ | **35.2** | **40.6** | **46.4** |

**Table 6.** Ablation experiment on the NWPU VHR-10 dataset.

| GSE | MPEAA | 3-Shot | 5-Shot | 10-Shot |
|:---:|:---:|:---:|:---:|:---:|
|  |  | 56.7 | 64.8 | 72.1 |
| √ |  | 59.7 | 67.6 | 73.6 |
|  | √ | 57.4 | 65.2 | 72.9 |
| √ | √ | **60.1** | **68.1** | **74.7** |

Figure 8 shows a visualization of the ablation experiment results. The baseline model cannot detect the targets of the two baseball fields at different sizes. After adding GSE, the model can detect ground track files with larger scales and smaller baseball files. However, our method can detect the baseball files with appearance changes and at minor scales after using the MPEAA module. This proves the effectiveness of our proposed method.

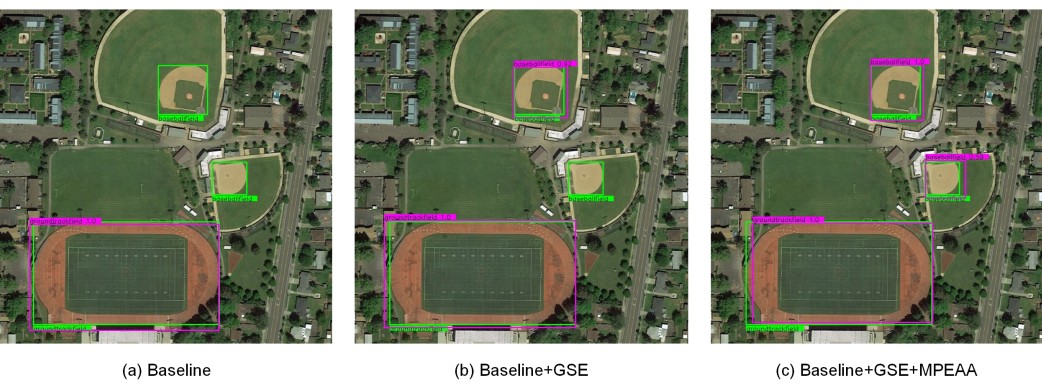

    (a) Baseline            (b) Baseline+GSE          (c) Baseline+GSE+MPEAA

**Figure 8.** Visualization of the ablation experiment results.

We compared the abilities of our method and the baseline model to detect the cross-scale targets. As shown in Figure 9, the baseline model cannot detect windmills with different scales. Thanks to the GSE strategy and the MPEAA module improving the ability of our method to detect multi-scale targets, our method can successfully detect windmills at each scale.

In addition, to explore the optimal number of the support image sizes in Gaussian-scale enhancement strategy, we conduct a comparison experiment setting the number of image sizes in the scale space from 2 to 5. We evaluate the indicators, including mAP and latency. Table 7 shows the detection accuracy with the 5-shot setting on the NWPU-VHR dataset and the training speed of each episode. When the number of support images' sizes is 5, the detector achieves the best accuracy of 68.2% mAP and the highest latency of 0.46 s. As observed, increasing the number of crop sizes will improve detection accuracy and reduce training speed. As the number of sizes increases, the range of learnable target scale distribution becomes wider. Constructing a complete scale space can effectively enhance scale information for network learning. Meanwhile, due to the increasing number of supporting images, it will inevitably take a longer time to train the network. Compared to size 2, size 4 improves accuracy by 1.3% mAP and increases the latency by 29%. Size 5

delivers 1.4% mAP improvement but increases the latency by 43%. The accuracy difference between the size 4 and size 5 is minimal, but the size 5 requires much more training time. Considering comprehensive accuracy and speed, cropping the support images into size 4 is a better choice.

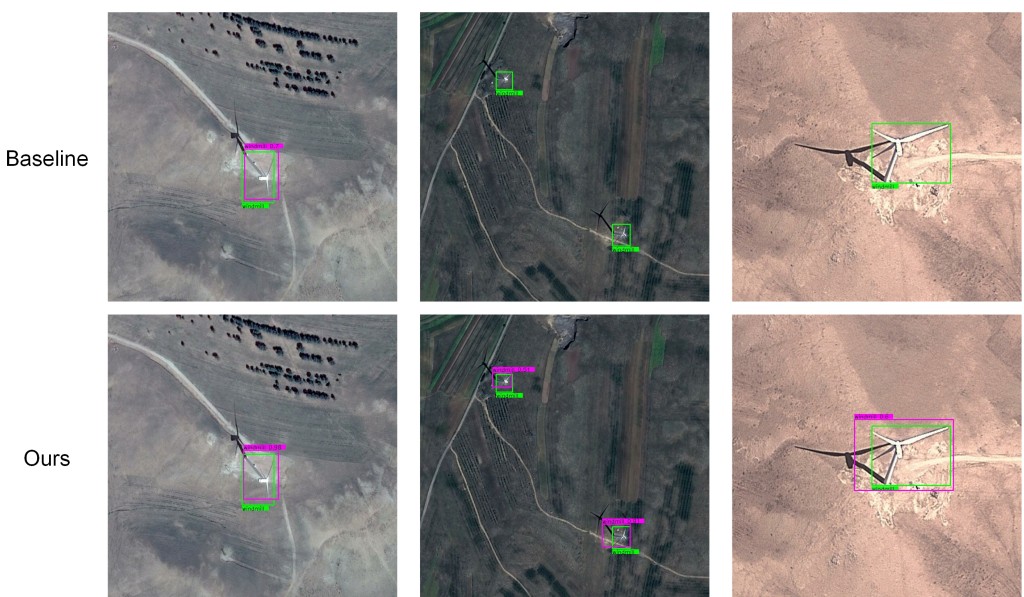

**Figure 9.** Comparison of windmill detection results with multi-scale size.

**Table 7.** Our comparison results with different numbers of the support image sizes.

| | $32^2$ | $64^2$ | $128^2$ | $256^2$ | $512^2$ | mAP (%) | Latency (s) |
|---|---|---|---|---|---|---|---|
| 2 sizes | | | | √ | √ | 66.8 | 0.326 |
| 3 sizes | | | √ | √ | √ | 67.6 | 0.375 |
| 4 sizes | | √ | √ | √ | √ | 68.1 | 0.423 |
| 5 sizes | √ | √ | √ | √ | √ | **68.2** | **0.468** |

## 4. Discussion

The effectiveness of our proposed Gaussian-scale augmentation strategy was also demonstrated by performing ablation experiments. We conducted experiments on the DIOR dataset and obtained the most significant improvement with the 5-shot setting, about 1.7%, compared with 1.3% with the 10-shot setting and 0.7% with the 20-shot setting. Our method improves more as the sample size decreases. Our method can provide more scale information since the scale information distribution (Figure 1) becomes sparser with fewer samples. A similar situation is observed on the NWPU VHR-10 dataset, with 3% improvement for 3 shots, 2.8% improvement for 5 shots, and 1.5% improvement for 10 shots, as expected.

## 5. Conclusions

In this paper, we proposed a new meta-learning-based few-shot object detection model to solve the problem of remote sensing images with cross-scale and sparse-scale information that exists when samples are scarce. According to the scale-space theory, we designed a Gaussian-scale enhancement strategy to enrich the scale information of the training samples by constructing a Gaussian-scale space for the support images. Furthermore, based on the Transformer's powerful self-attention and context acquisition capabilities, we designed a multi-branch patch-embedding attention aggregation module, which can better learn multi-scale features to improve detection accuracy. We conducted comparison and ablation experiments on two datasets to demonstrate the effectiveness of our method.



**Author Contributions:** Conceptualization, Z.Y. (Zhibin Yu) and Z.Y. (Zhenyu Yang); methodology, Z.Y. (Zhenyu Yang) and Z.Y. (Zhibin Yu); writing—original draft preparation, Z.Y. (Zhenyu Yang) and Y.Z.; writing—review and editing, Z.Y. (Zhibin Yu); software, Z.Y. (Zhibin Yu) and J.Z.; visualization, Y.Z. and J.Z.; supervision, B.Z.; project administration, B.Z.; funding acquisition, Z.Y. (Zhibin Yu). All authors have read and agreed to the published version of the manuscript.

**Funding:** This work was supported by the Natural Science Foundation of Shandong Province of China under grant number ZR2021LZH005; by the National Natural Science Foundation of China under grant number 62171419; and by Hainan Province Science and Technology Special Fund, China (ZDYF2022SHFZ318).

**Data Availability Statement:** The experiments are evaluated on publicly open datasets. The datasets can be accessed in their corresponding published papers.

**Conflicts of Interest:** The authors declare no conflict of interest.

## Abbreviations

The following abbreviations are used in this manuscript:

| | |
|---|---|
| GSE | Gaussian-scale enhancement |
| MPEAA | Multi-branch patch-embedding attention aggregation |
| OBIA | Object-based image analysis |
| R-CNN | Region-based convolutional neural networks |
| FPN | Feature pyramid networks |
| DeFRCN | Decoupled Faster R-CNN |
| RPN | Region proposal network |
| RoI | Region of interest |
| SIFT | Scale-invariant feature transform |
| MSFA | Multi-level support feature aggregation |
| Concat | Concatenate |
| ViT | Vision Transformer |
| SGD | Stochastic gradient descent |
| mAP | Mean average precision |

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
