# Peer review of "Scale Information Enhancement for Few-Shot Object Detection on Remote Sensing Images"

_remotesensing, doi:10.3390/rs15225372_

Round 1
Reviewer 1 Report
Comments and Suggestions for Authors
Summary
This paper proposes to design a Gaussian scale enhancement (GSE) strategy and a multi-scale patch embedding attention aggregation (MPEAA) module for a few-shot object detection task. Specifically, the GSE constructs Gaussian scale space for small targets. Furthermore, the MPEAA uses meta-learning to learn multi-scale features and improve the performance of detecting new classes. The experiments show that the proposed method outperforms other state-of-the-art baselines on several datasets.
Strengths
1.The paper presents a new method of applying the few-shot techniques to remote-sensing images. The model can enrich the scale information of the object and learn better multi-scale features to detect the remote sensing images.
2.Good set of experiments for yielding state-of-the-art results on several datasets.
Weaknesses
1. The analysis of the Gaussian scale strategy is not sufficient. The authors crop the images into four sizes, why the number is four? What if the number is three or two?
2. I can not get the novelty of the MPEAA. Since the other works also use a cross-attention mechanism, it seems that the authors match the different scale patches and the other parts are the same as the previous work. The authors should strengthen your highlight.
3. Writing can be improved, the article typesetting should be corrected. In addition, the papers struggle to highlight the experiment results and the authors need to readjust the results tables.
Comments on the Quality of English LanguageThe motivation of this paper is not clear. The authors should modify this paper carefully.
Author Response
Reviewer1
- The analysis of the Gaussian scale strategy is not sufficient. The authors crop the images into four sizes, why the number is four? What if the number is three or two?
Response:
Thanks for your constructive suggestions. We conduct a comparison experiment on selecting the numbers of support images' sizes on the NWPU-VHR dataset. We set the number of image sizes in the scale space from 2 to 5. We evaluate the indicators, including mAP and latency. Table 1 shows the detection accuracy of novel classes under the 5-shot setting and the training speed of each episode. When the number of support images' sizes is 5, the detector achieves the best accuracy of 68.2 \% mAP and the highest latency of 0.46s. As observed, increasing the number of crop sizes will improve detection accuracy and reduce training speed. We analyze that as the number of sizes increases, the range of learnable target scale distribution becomes wider. Constructing a complete scale space can effectively enhance scale information for network learning.
Meanwhile, due to the increasing number of supporting images, it will inevitably take a longer time to train the network. Compared to size 2, size 4 improves accuracy by 1.3\% mAP and increases the latency by 29\%. Size 5 delivers 1.4\% mAP improvement but increases the latency by 43\%. The accuracy difference between the size 4 and 5 is minimal, but the size 5 requires much more training time. Considering comprehensive accuracy and speed, cropping the support images into size 4 is a better choice.
Finally we add experimental results and discussion of selecting the numbers of support images' sizes in Sec. 3.4 "Abalation Experiment" (in Page 15-16 Line 419-434, text in cyan color) as follows:
``In addition, to explore the optimal number of the support image sizes in Gaussian-scale enhancement strategy, we conduct a comparison experiment setting the number of image sizes in the scale space from 2 to 5. We evaluate the indicators, including mAP and latency. Table 1shows the detection accuracy with the 5-shot setting on the NWPU-VHR dataset and the training speed of each episode. When the number of support images' sizes is 5, the detector achieves the best accuracy of 68.2 \% mAP and the highest latency of 0.46s. As observed, increasing the number of crop sizes will improve detection accuracy and reduce training speed. As the number of sizes increases, the range of learnable target scale distribution becomes wider. Constructing a complete scale space can effectively enhance scale information for network learning. Meanwhile, due to the increasing number of supporting images, it will inevitably take a longer time to train the network. Compared to size 2, size 4 improves accuracy by 1.3\% mAP and increases the latency by 29\%. Size 5 delivers 1.4\% mAP improvement but increases the latency by 43\%. The accuracy difference between the size 4 and size 5 is minimal, but the size 5 requires much more training time. Considering comprehensive accuracy and speed, cropping the support images into size 4 is a better choice. ''
- I can not get the novelty of the MPEAA. Since the other works also use a cross-attention mechanism, it seems that the authors match the different scale patches and the other parts are the same as the previous work. The authors should strengthen your highlight.
Response:
Thanks for your constructive suggestions. We re-summarized the innovation point of the MPEAA module. The MPEAA module uses multi-scale patch embbedding and multi-branch aggregation structure. In previous meta-learning networks, support and query features only interacted in a single manner, which did not fully utilize multi-scale information. We use multi-scale patch embedding to represent fine and coarse features effectively. Multi-branchs perform feature re-weighting based on Transformer attention, and the resulting features are aggregated, which can better learn multi-scale features to alleviate the cross-scale problem.
We have re-written Sec. 1.3 "Problems and Contribution" (in Page 4 Line 146-151, text in cyan color) as follows:
``We propose a multi-branch patch-embedding attention aggregation module with meta-learning. We use multi-scale patch embedding to represent fine and coarse features effectively. Multi-branchs perform feature re-weighting based on Transformer attention. The resulting features are aggregated, which can better learn multi-scale features to alleviate the cross-scale problem and improve the performance of detecting new classes of targets. ''
- Writing can be improved, the article typesetting should be corrected. In addition, the papers struggle to highlight the experiment results and the authors need to readjust the results tables.
Response:
Thanks for your constructive suggestions. We have tried our best to correct the grammar mistakes and invite the native speaker for proofreading. As shown in Table 2, we highlight the mAP score of the best detection performance in bold for Table 2-7.
We have re-written Sec. 3.3 "Results" (in Page 11 Line 363-366, text in cyan color) as follows:
``We highlight the mAP score of the best detection performance in bold for Table 2-7. Compared with the baseline model DCNet, our method has a 1.5\% improvement in the 20-shot setting, a 2.3\% improvement in the 10-shot setting, and a 2\% improvement in the 5-shot setting. ''

Reviewer 2 Report
Comments and Suggestions for Authors
Inadequate information could lead to unbalanced representations of various item scales. Improving scales via a Gaussian approach might make this mismatch worse and lead to forecasts that are biased in favor of the artificially improved scales.
It is assumed that objects of interest in gaussian scale space are smooth and experience moderate variations in intensity. It might not be able to recognize things with incredibly abrupt or irregular patterns, which would limit its ability to adapt to different item appearances.
Rich context information could be lost when feature maps are directly combined after being upsampled. When employing multi-scale patch embedding attention aggregation, this can be a drawback because the model might not be able to extract all the relevant information from the image.
In Line 270 Depending on the size of the objects inside the patches, overlapping patches could behave differently. It is possible that smaller objects do not gain as much from overlaps, which could cause differences in detection or segmentation performance based on object scales.
Author Response
Reviewer 2 Comment 1
Inadequate information could lead to unbalanced representations of various item scales. Improving scales via a Gaussian approach might make this mismatch worse and lead to forecasts that are biased in favor of the artificially improved scales.
It is assumed that objects of interest in gaussian scale space are smooth and experience moderate variations in intensity. It might not be able to recognize things with incredibly abrupt or irregular patterns, which would limit its ability to adapt to different item appearances.
Response:
Thanks for your constructive suggestions. Following the scale-space theory [A][B][C], we can construct a scale space by convolving the target object image with a series of Gaussian filters with different parameters. Although the GSE strategy we proposed cannot make the scale information completely consistent with the real distribution, it can increase scale information and alleviate the problem of sparse scale distribution. Our visualization results in Figure 8 also prove that our method is better at detecting targets of multiple scales. In contrast, the baseline method can only detect targets of a single scale. Meanwhile, by using the GSE strategy, our method has achieved about 2-3% mAP improvement in accuracy on DIOR and NWPU-VHR datasets. It proves the effectiveness of using Gaussian convolution to rich scale information.
Our GSE strategy was not designed for incredibly abrupt or irregular patterns detection. However, the MPEAA module uses a global attention mechanism for pixel-level matching that can mitigate the problem of appearance changes. The helicopter can be considered as a class with similar patterns of planes. The proposed model can successfully recognize the helicopter with the help of MPEAA module.
[A] Witkin, A. Scale-space filtering: A new approach to multi-scale description. In Proceedings
of the ICASSP’84. IEEE International Conference on Acoustics, Speech, and Signal
Processing. IEEE, 1984, Vol. 9, pp. 150-153.
[B] Lindeberg, T. Scale-space theory: A basic tool for analyzing structures at different scales.
Journal of applied statistics 1994, 21, 225-270.
[C] Lowe, D.G. Object recognition from local scale-invariant features. In Proceedings of the Proceedings of the seventh IEEE international conference on computer vision. Ieee, 1999, Vol. 2, pp.1150-1157.
We have re-written Sec. 2.2 "Gaussian-scale enhancement strategy" (in Page 8 Line 247-249, text in cyan color) as follows:
“ Although the GSE strategy we proposed cannot make the scale information completely consistent with the real distribution, it can increase scale information and alleviate the problem of sparse scale distribution. ”
Reviewer 2 Comment 2
Rich context information could be lost when feature maps are directly combined after being upsampled. When employing multi-scale patch embedding attention aggregation,this can be a drawback because the model might not be able to extract all the relevant information from the image.
Response:
Thanks for your constructive suggestions. In the multi-branch patch embedding attention aggregation module, we adopt the method of gradually up-sampling and combining features of different scales to integrate multi-scale features better.
[A] and [B] mentioned that aligning and combining features of different scales can achieve multi-scale feature fusion well. Based on this idea, we use up-sampling to align the spatial size of multi-scale features and perform element-wise additions to integrate features. In the case of limited samples, it is difficult for the network to learn and utilize all information thoroughly. Although our model might not extract all the relevant information from the image, the MPEAA module use multi-scale patch embedding to obtain both fine and coarse feature representations, which can gain more information and enable the network to learn multi-scale features better. As shown in the ablation experiment results in Tables 5 and 6, the MPEAA module improves the performance of detecting targets.
[A] Lin T Y, Dollár P, Girshick R, et al. Feature pyramid networks for object detection[C]// Proceedings of the IEEE conference on computer vision and pattern recognition. 2017: 21172125.
[B] Zhang G, Li Z, Li J, et al. Cfnet: Cascade fusion network for dense prediction[J]. arxiv preprint arxiv:2302.06052, 2023.
We have re-written Sec. 2.3 "Multi-branch patch-embedding attention aggregation" (in Page 9 Line 294-296, text in cyan color) as follows:
“ We adopt the method of gradually upsampling and combining features of different scales, which can integrate multi-scale features well. We additionally concatenate multi-branch fusion features and the query feature to keep the information intact. ”
Reviewer 2 Comment 3
In Line 270 Depending on the size of the objects inside the patches, overlapping patches could behave differently. It is possible that smaller objects do not gain as much from overlaps, which could cause differences in detection or segmentation performance based on object scales.
Response:
Thanks for your constructive suggestions. Overlapping patch embedding methods have little effect on the performance of small object detection. Overlapping patch embedding is proposed in PVTV2 [A]. [A] proves that overlapping patch embedding performs better on classification and detection tasks than original patch embedding because it can model the local continuity of images and feature maps via the overlapping sliding window. Thus, we use overlapping patch embedding on the MPEAA module to divide the images into several patches. Although overlapping patch embedding may not improve performance for detecting small targets, the GSE strategy can increase the scale information and diversity of small targets, which helps detect small targets
[A] Wang W, Xie E, Li X, et al. Pvt v2: Improved baselines with pyramid vision transformer[J]. Computational Visual Media, 2022, 8(3): 415-424. We have re-written Sec. 2.3 "Multi-branch patch-embedding attention aggregation" (in Page 8-9 Line 267-271, text in cyan color) as follows:
“ Overlapping patch embedding is proposed in PVTV2 [69]. [69] proves that overlapping patch embedding performs better on classification and detection tasks than original patch embedding because it can model the local continuity of images and feature maps via the overlapping sliding window. So, we use overlapping patch embedding to divide the images into several patches on the MPEAA module. ”
Please see the attachment for detail.

Reviewer 3 Report
Comments and Suggestions for Authors
Dear Authors,
The Manuscript describes solution of scale invariant object detection in remote sensing images using GSE and MPEAA methods. This problem is very important for many applications.
Specific comments:
11) Introduction includes presentation of results (Figure 1) and detailing methodology on page 2. It is suggested to move these parts into relevant sections.
22) Section 2 has to be a part of introduction.
33) Figure 3 shows a block with Gaussian Blur function. Please explain its importance in text.
44) Page 7, Line 236: What scale parameters σ were used in Gaussian convolution kernels?
55) In description of datasets (Section 4.1) please specify if they are based on satellite or airborne, including UAV, imagery.
66) Page 10. The proposed detector was compared to different methods. Please add references to each method. Did Authors compare with methods described in [58] and [59]?
77) Please provide test results on both satellite and airborne imagery. Authors can use Google Earh, Bing and other freely available data sources. It will allow to understand efficiency and practical importance of the results.
Sincerely,
Reviewer
Comments on the Quality of English LanguageEnglish proofreading/spelling check is required.
Author Response
Reviewer 3
Reviewer 3 Comment 1
- Introduction includes presentation of results (Figure 1) and detailing methodology on page 2. It is suggested to move these parts into relevant sections.
Response: Thanks for your constructive suggestions. First, we have moved Figure 1 to Sec. 3.4 "Abalation Experiment" (in Page 6). We have taken this figure as a presentation of ablation experimental results and provided a more precise diagram explanation. Compared to the baseline method, our method can successfully detect targets with different scales. The figure shows that after using the GSE strategy and the MPEAA module, our method improves the ability to detect cross-scale objects.
We have re-written Sec. 3.4 "Abalation Experiment" (in Page 15 Line 414-418, text in cyan color) as follows:
“We compared the abilities of our method and the baseline model to detect the cross-scale targets. The baseline model cannot detect windmills with different scales. Thanks to the GSE strategy and the MPEAA module improving the ability of our method to detect multi-scale targets, our method can successfully detect windmills at each scale.”
Second, we have moved detailing methodology on page 2 to Method (Section 2). We have re-written Sec. 2 "Method" (in Page 5 Line 165-168, text in cyan color) as follows:
“Query features use three different sizes of patch embeddings for encoding. The output of the query branch features patch embeddings of different sizes. We used the self-attention method in the Transformer for re-weighting features and then fused the multi-branch outputs in a pyramidal manner to obtain a feature map containing multi-scale information. ”
Finally we have re-written Sec. 1.3 "Problems and Contribution" (in Page 3 Line 134-140, text in cyan color) as follows:
“To address these problems, we propose a few-shot object detection method on remote sensing images, design a Gaussian-scale enhancement strategy, and add a multi-scale attention module. To tackle the sparsity problem of scale distribution with few training samples, we proposed the Gaussian-scale enhancement strategy, which can enrich the scale information of the target. In the MPEAA module, we use multi-scale patch embedding to obtain the feature map of different receptive fields and calculate the attention. Furthermore, the multi-path aggregation method we designed can learn better multi-scale features. ”
Reviewer 3 Comment 2
- Section 2 has to be a part of introduction.
Response: Thanks for your valuable suggestions. We have reorganized the structure of Sec. 1 "Introduction" to provide better readability. We have moved related work to the introduction and divided the introduction into three sections. First, Sec. 1.1 "Background" mainly introduces
the current research background and the significance of studying few-shot object detection on remote sensing images. Then, Sec. 1.2 "Related Work" describes the research closely related to this article, including remote sensing image object detection, few-shot object detection, and
few-shot object detection on remote sensing images. Finally, we introduce the cross-scale and sparse scale distribution problem with limited remote sensing images in Sec. 1.3 "Problems and Contribution". We also list the contribution of our paper to address theses problems.
Reviewer 3 Comment 3
- Figure 3 shows a block with Gaussian Blur function. Please explain its importance in text.
Response: Thanks for your constructive suggestions. Following the scale-space theory, we can construct a scale space by convolving the target object image with a series of Gaussian filters with different parameters.
We have re-written Sec. 2 "Method" (in Page 5 Figure 2, text in cyan color) as follows:
“ ... The role of the Gaussian Convolution module is to construct a Gaussian scale-space and enrich the scale information on the support image. ... ”
Reviewer 3 Comment 4
- Page 7, Line 236: What scale parameters σ were used in Gaussian convolution kernels?
Response: Thanks for your constructive suggestions. σ is an important parameter describing the degree of Gaussian blur. The larger the σ, the greater the degree of blur. The way we select the σ parameter here is to use the calculation formula provided by the image processing library
OpenCV. The σ calculation of this formula is related to the size of the convolution kernel. We take the size of the convolution kernel from small to large {3, 5, 7, 9}, thereby calculating the size of the σ parameter as.
We have re-written Sec. 2.2 "Gaussian scale enhancement strategy" (in Page 6-7 Line 228-231, text in cyan color) as follows:
“Image processing library OpenCV provides a formula 4 for the adaptive calculation of σ. ksize represents the size of the convolution kernel, which is taken as odd numbers. We take ksize with 3, 5, 7, and 9, respectively, and get the value of σi = {0.8, 1.1, 1.4, 1.7}. ”
Reviewer 3 Comment 5
- In description of datasets (Section 4.1) please specify if they are based on satellite or airborne, including UAV, imagery.
Response: Thanks for your detailed suggestions. We investigated the sources of the experimental datasets. The datasets DIOR and NWPU-VHR are satellite images from Google Earth. We additionally use DOTA, an aerial image dataset, for experiments to verify the effectiveness of our proposed method.
we have re-written Sec. 3.1 "Datasets" (in Page 9-10 Line 302-304 and Page 10 Line 307-308, text in cyan color) as follows:
“ The NWPU VHR-10 [53-55] dataset contains 800 high-resolution satellite images collected from Google Earth and Vaihingen datasets and then manually annotated by experts. ”
“ ...The DIOR dataset are satellite images from Google Earth and containing 23463 images and 192472 instances, covering 20 object classes. ”
Reviewer 3 Comment 6
6. Page 10. The proposed detector was compared to different methods. Please add references to each method. Did Authors compare with methods described in [58] and [59]?
Response: Thanks for your constructive suggestions. First, we added literature quotation in the article.
We have re-written Sec. 3.3 "Results" (in Page 11 Line 354-361, text in cyan color) as follows:
“We validated our proposed few-shot detector on the NWPU VHR-10 and DIOR datasets along with state-of-the-art FSOD methods, such as meta-learning-based Rep- Met [40], FSODM [50], PAMS-Det [51], DCNet [43], FCT [44] and fine-tuning-based CIR-FSD [52], MPSR [47] ,TFA [46], SAGS [29], and DeFRCN [48]. The meta-learning- based methods construct several episodes to learn and obtain the class representative
vector. These methods then use the class representative vector to re-weight the query features and to predict the target. The transfer-learning-based methods use a batch training strategy and then directly predict the target in the image. ”
Meanwhile, RepMet [58] and TFA [59] are classical few-shot object detection algorithms which have been experimented with in previous work [52]. We directly transferred the experimental results in [52] and compared our method with them. We have added the detection results of method RepMet [58] and TFA [59].
Reviewer 3 Comment 7
- Please provide test results on both satellite and airborne imagery. Authors can useGoogle Earh, Bing and other freely available data sources. It will allow to understandefficiency and practical importance of the results.
Response: Thanks for your constructive suggestions. Our previous experiments were mainly conducted on satellite remote-sensing images, including DIOR and NWPU VHR-10 datasets. We have found an additional aerial image dataset, DOTA, and conducted comparative experiments with various methods. The DOTA dataset, which stands for "Dataset for Object Detection in Aerial Images," is specifically designed for detecting and classifying various object categories within high-resolution aerial images. Aerial imagery is collected using aircraft or drones equipped with high-resolution cameras. Currently, there are multiple versions of DOTA datasets. This paper uses the DOTAv1.0 version for the study of few-shot object detection. The DOTA dataset contains 2806 images of 15 classes. The object classes include planes, ships, storage tanks, baseball diamonds, tennis courts, basketball courts, ground-track fields, harbors, bridges, large vehicles, small vehicles, helicopters, roundabouts, soccer ball fields and swimming pools. Four classes (planes, baseball fields, tennis courts and helicopters) were used as novel classes, and the others were used as base classes. For the DOTA dataset, the number of annotated instances K was set as 3, 5, 10, and 20. Compared to other few-shot object detectors, our method has obvious advantages in detection accuracy with each shot setting. The visual results also show that our proposed GSE strategy and MPEAA module can improve the model’s ability to detect multi-scale and densely arranged targets. Our method can correctly detect targets even if the plane and helicopter have similar characteristics.
We have re-written Sec. 3.1 "Datasets" (in Page 10 Line 312-320, Page 10 Line 326-327 and Page 10 Line 332-333 , text in cyan color) as follows:
“ The DOTA ("Dataset for Object Detection in Aerial Images") dataset is specifically designed for detecting and classifying various object categories within high-resolution aerial images. Aerial imagery is collected using aircraft or drones equipped with high-resolution cameras. Currently, there are multiple versions of DOTA datasets. This paper uses the DOTAv1.0 version for the study of few-shot object detection. The DOTA dataset contains 2806 images of 15 classes. The object classes include planes, ships, storage tanks, baseball diamonds, tennis courts, basketball courts, ground track fields, harbors, bridges, large vehicles, small vehicles, helicopters, roundabouts, soccer ball fields and swimming pools. ”
“ ... For the aerial dataset DOTA, we selected four novel classes (plane, baseball-field, tennis-court and helicopter). ”
“ ... For the DOTA dataset, the number of annotated instances K was set as 3, 5, 10, and 20. ... ”
Our method consistently outperforms DCNet and other counterparts with each shot setting. Compared to MPSR, our method significantly improves accuracy by 1.7%/3.8%/2.9%/3.6% mAP with 3/5/10/20-shot setting. The detection result verifies the effectiveness of our proposed method on satellite and aerial image data. ”
With the help of the GSE strategy, our method can successfully detect planes and baseball fields of various sizes, and other methods will lose some targets. Our proposed MPEAA module can achieve high-quality feature reweighting, and our method can detect densely arranged tennis courts. The red box represents a misdetection. The helicopter can be considered as a class with similar patterns of planes, causing both MPSR and DCNet to detect helicopters as airplanes. The proposed model can successfully recognize the helicopter with the help of MPEAA module. ”
Please see the attachment for detail.
